# Mechanism of Plantamajoside in inhibiting ferroptosis of pancreatic β cells and treatment of T2DM via activation of the xCT/GPX4 pathway

Hongmin Zhao[1‡], Renlin Li[2‡], Xuan Guo[1], Jingrui Kang[1], Huajun Li[1], Xiaoyun Wang[1], Yuansong Wang[1], Huantian Cui[2], Shuquan Lv[1*], Weibo Wen🔘[2*], Zhongyong Zhang[1*]

**1** Cangzhou Hospital of Integrated Traditional Chinese Medicine and Western Medicine of Hebei, Hebei, China, **2** Yunnan University of Chinese Medicine, Kunming, China

‡ These authors share first authorship on this work.
* czlvshuquan@163.com (SL); wenweibo2020@163.com (WW); zzyhappy666@126.com (ZZ)

## Abstract

Pancreatic β-cell damage, a key pathology in Type 2 Diabetes Mellitus (T2DM), may be mitigated by inhibiting ferroptosis. Plantamajoside (PMS) shows promise in alleviating cellular damage and improving T2DM outcomes, though its mechanisms remain unclear. This study investigated PMS's role in suppressing ferroptosis in pancreatic β-cells via the cysteine/glutamate transporter (xCT)/ glutathione peroxidase 4 (GPX4) pathway. In our *in vivo* experiments, PMS was administered to T2DM mice via gavage, and its effects on tissue damage, ferroptosis, and xCT/GPX4 pathway modulation were assessed. Furthermore, *in vitro* experiments employed high glucose (HG) and palmitic acid (PA) conditions, to induce damage in pancreatic β-cells. We investigated the beneficial impacts of PMS on pancreatic β-cell damage, its modulation of ferroptosis, and its influence on the xCT/GPX4 pathway. To compare the capacity of PMS to inhibit ferroptosis, we utilized the ferroptosis inhibitor ferrostatin-1 (Fer-1) as a positive control, while the GPX4 inhibitor RSL-3 validated PMS's mechanism through the xCT/GPX4 axis. Our findings revealed that PMS effectively mitigated pancreatic tissue damage in T2DM mice, reduced ferroptosis, and enhanced the expression of factors associated with the xCT/GPX4 pathway. Moreover, PMS alleviated HG and PA-induced damage in pancreatic β-cells, suppressed ferroptosis, and upregulated factors linked to the xCT/GPX4 pathway. Similar to the ferroptosis inhibitor Fer-1, PMS exhibited comparable effects. Conversely, RSL-3 attenuated the protective effects of PMS on pancreatic β-cell damage, its inhibition of ferroptosis, and its activation of the xCT/GPX4 pathway. PMS exhibited the capacity to diminish damage to pancreatic islet β-cells induced by T2DM, both in *vivo* and in *vitro*. This favorable outcome may stem from the alleviation of lipid peroxidation and reduction of ferroptosis. Moreover, this regulatory mechanism was accomplished through the enhancement of the xCT/GPX4 axis.

**Data availability statement:** All relevant data are within the manuscript and its Supporting Information files.

**Funding:** This work was supported by the Yuansong Wang National Famous Traditional Chinese Medicine Expert Heritage Studio (grant no. 4 [2022]) and Science and Technology Program of Yunnan Province (202301AZ070001-011).

**Competing interests:** The authors have declared that no competing interests exist.

**Abbreviations:** T2DM: Type 2 Diabetes Mellitus; PMS: Plantamajoside; HG: glucose; PA: palmitic acid; Xct: cysteine/glutamate transporter; GPX4: glutathione peroxidase 4; Fer-1: ferrostatin-1; MDA: malondialdehyde; ROS: reactive oxygen species; 4-HNE: 4-hydroxynonenal; GSH: glutathione; GSSG: glutathione disulfide; SLC7A11: solute carrier family 7A11; TRF: Transferrin; SLC3A2: solute carrier family 3 member 2; FTL: ferritin light chain; ACSL4:Long-chain acyl-coenzyme A (CoA) synthase 4; AOD: Average Optical Density.

## 1. Introduction

Type 2 diabetes mellitus (T2DM) presents as a chronic metabolic disorder marked by insulin resistance and inadequate pancreatic β-cell function. It constitutes over 90% of diabetes cases globally and stands as a significant challenge for public health worldwide [1]. The onset of T2DM involves a complex interplay of genetic predisposition, environmental factors, and lifestyle choices. Recent investigations underscore the progressive decline in pancreatic β-cell function and diminished insulin sensitivity as central elements in T2DM pathogenesis [2]. Hence, early detection and intervention targeting pancreatic β-cell dysfunction and insulin resistance hold paramount importance for T2DM prevention and management. Despite the array of pharmaceuticals and treatment modalities available for T2DM control, many patients grapple with attaining optimal blood glucose levels, with long-term medication posing risks of adverse effects and complications [3]. The search for more efficacious and safer treatment avenues remains a pivotal pursuit in diabetes research.

Ferroptosis is a distinct form of cell death instigated by iron-dependent lipid peroxidation. It manifests through distinct biochemical characteristics, including iron ion accumulation, substantial generation of lipid peroxides, and depletion of glutathione [4]. Emerging evidence posits ferroptosis as a pivotal player in pancreatic β-cell injury and T2DM pathogenesis [5]. Within the pathological milieu of T2DM, chronic hyperglycemia fosters heightened oxidative stress, thus influencing iron metabolism and lipid peroxidation levels, ultimately instigating ferroptosis [6]. The heightened presence of iron ions amplifies lipid peroxide levels within pancreatic β-cells, thereby exacerbating ferroptosis and aggravating pancreatic β-cell damage [7]. Attenuating ferroptosis in pancreatic β-cells is a promising therapeutic avenue for T2DM. Metformin (MET) is validated for its therapeutic efficacy in T2DM by virtue of its inhibition of ferroptosis in pancreatic β-cells [8].

Plantamajoside (PMS, $C_{29}H_{36}O_{16}$), a major ingredient isolated from *Plantago asiatica L.* (Plantaginaceae) and identified as a unique phenylpropanoid glycoside in Herba plantaginis, exhibits notable anti-inflammatory, antioxidant, and anti-apoptotic properties, making it a subject of scientific interest [9]. Studies have emphasized the capacity of PMS to safeguard cells against cellular injury and stress, such as oxidative stress and inflammation [10], and alleviate high glucose-induced damage in rat glomerular mesangial cells [11]. However, its therapeutic potential and underlying mechanism in T2DM remain poorly understood. Recent studies highlight that the cysteine/glutamate transporter (XCT)/glutathione peroxidase 4 (GPX4) pathway is a critical defense mechanism against ferroptosis, particularly in pancreatic β-cells [12]. XCT (SLC7A11/SLC3A2), a cystine/glutamate antiporter, mediates cystine uptake to support glutathione (GSH) synthesis [13]. GPX4, in turn, utilizes GSH to neutralize lipid peroxides, thereby suppressing oxidative stress and ferroptosis [14]. Given the established role of ferroptosis in β-cell dysfunction and the limited understanding of PMS's therapeutic mechanisms in T2DM, this study aims to explore whether PMS can ameliorate T2DM by modulating the XCT/GPX4 pathway to inhibit ferroptosis in pancreatic β-cells.

In our *vivo* experiments, we established a T2DM mouse model and orally administered PMS to explore its therapeutic efficacy and its impact on pancreatic tissue ferroptosis, alongside evaluating the expression levels of factors linked to the xCT/GPX4 pathway. Moreover, our in *vitro* experiments employed a combination of high glucose (HG) and palmitic acid (PA) to induce pancreatic β-cell injury, probing the protective effects of PMS on pancreatic β-cells and its influence on ferroptosis and the xCT/GPX4 pathway. We utilized ferrostatin-1 (Fer-1), a ferroptosis inhibitor, as a positive control to compare the efficacy of PMS in ferroptosis inhibition and employed RSL-3, a GPX4 inhibitor, to validate whether PMS ameliorates ferroptosis in pancreatic β-cells via the xCT/GPX4 pathway.

## 2. Method

### 2.1 Reagent

Extensive details pertaining to reagents, kits, and antibodies are provided within the supplementary materials.

### 2.2 *In vivo* study

**Model establishment and dosing protocol.** C57BL/6 mice were procured from SPF (Beijing) Biotechnology Co., Ltd (SCXK (Beijing) 2019−0010). and housed in specific pathogen-free (SPF) conditions. All experimental procedures followed the Guidelines for Animal Ethics and received approval from Cangzhou Hospital of Integrated Traditional Chinese Medicine and Western Medicine of Hebei Province (Approval Number: CZX2024-KY-105).

To induce the T2DM, we utilized the HFD + STZ method as previously outlined. Briefly, mice were fed a high-fat, high-sugar diet (HFD) for eight weeks. At the end of the 8th week, we administered a single intraperitoneal injection of STZ (Streptozotocin) at a dose of 30 mg/kg. Following this injection, the mice continued on the HFD until the conclusion of the experiment. Successful establishment of the T2DM model was confirmed by a random blood glucose level of ≥16.7 mmol/L [15].

We randomly allocated a total of 60 mice into six groups: the control group (CON), T2DM model group (T2DM), the positive control group (PC), the PMS low-dose group (PMSL), the PMS medium-dose group (PMSM), and the PMS high-dose group (PMSH). Except for the Con group, all other groups were induced to develop T2DM. Following successful establishment of the models, the PC group received a daily oral administration of 250 mg/kg/d of MET, while the PMSL, PMSM, and PMSH groups received daily oral administrations of 25 mg/kg/d, 50 mg/kg/d, and 100 mg/kg/d of PMS, respectively. The dosage of PMS were set according to previous studies [16–18]. The Con and T2DM groups received the same volume of vehicle daily. This intervention period spanned eight weeks, during which we monitored the body weight and fasting blood glucose (FBG) levels of the mice in each group on a weekly basis.

After eight weeks of treatment, we collected 24-hour urine samples from each group using metabolic cages. Following anesthesia with sodium pentobarbital (50 mg/kg), blood samples were collected from the abdominal aorta, centrifuged to obtain the supernatant, and stored at −80 °C. Subsequently, animals were euthanized by cervical dislocation, pancreatic tissue samples were collected, with a portion fixed in formalin solution and the remainder stored in cryovials at −80 °C for further analysis.

**Oral glucose tolerance test.** After eight weeks of administering the drugs, we conducted an oral glucose tolerance test (OGTT). Mice were fasted for 12 hours with access to water. We measured their FBG levels. Following this, we administered a 50% glucose solution at a dose of 2 g·kg$^{-1}$. We collected blood samples from the tail vein at 15 min, 30 min, 60 min, 90 min, and 120 min intervals to measure blood glucose levels. Subsequently, we plotted an OGTT curve and calculated the area under the curve (AUC) for glycemia to evaluate glucose tolerance for each group of mice.

**HbA1c measurement.** Following the completion of modeling and drug administration, we anesthetized the mice and collected blood from the abdominal aorta. After centrifugation at 1000 × g for 10 minutes, we collected the upper serum layer for further analysis. The serum HbA1c (Hemoglobin A1c) levels of mice in each group were measured according to the instructions provided with the HbA1c kit.

**HOMA-IR measurement.** We collected serum samples from mice in each group and measured the concentration of fasting insulin (FINS) using an ELISA kit. Subsequently, we calculated the insulin resistance index using the HOMA-IR (Homeostatic Model Assessment of Insulin Resistance) formula: HOMA-IR = [Fasting Insulin (μU/mL) × FBG (mmol/L)]/ 22.5.

**H&E staining.** We harvested pancreatic tissues from mice in each group. Half of the tissues were preserved at −80 °C for subsequent analysis, while the remaining half were immersed in 4% paraformaldehyde for 24 hours. After dehydration with ethanol, embed the tissue in paraffin and cut into 5μm sections. Perform H&E (Hematoxylin and Eosin) staining according to previous studies [19]. Ultimately, we observed and photographed the morphology of the pancreatic islets under a microscope.

**TUNEL staining.** TUNEL staining was performed on pancreatic tissue to assess cell apoptosis, as described in previous studies [20]. These stained sections were observed under a fluorescence microscope. The rate of TUNEL-positive cells in the islet was analyzed using Image-Pro Plus 6.0 software.

**Perls Prussian blue staining.** Perls' staining was performed on paraffin-embedded pancreatic tissue sections to assess iron deposition, as described in previous studies [21]. Briefly, the slides were incubated for 15min in iron stain solution. Nuclear Fast Red was used as counterstain. Finally, we observed iron deposition in the pancreatic tissue using an optical microscope.

## 2.3 *In vitro* study

**Cell culture.** We cultured Min6 cells in 1640 medium supplemented with 10% fetal bovine serum (FBS), 100 μg/mL streptomycin, and 100 U/mL penicillin. The cells were kept in a humidified environment at 37 °C with 5% $CO_2$ and 95% air. We replaced the medium with fresh one every two days and performed cell passaging when 80% confluence was reached. Logarithmic growth phase cells were utilized for experiments.

**Cell modeling and grouping.** We conducted the cytotoxicity assay of Min6 cells using the MTT assay. Min6 cells were seeded into a 96-well plate at a density of $1 \times 10^4$ cells/well and incubated for 24 hours. After incubation, we divided the cells randomly into a blank control group and different concentration gradients of PMS (0, 12.5, 25, 50, 100, 200 μmol/L) treatment groups, with six replicate wells in each group. Following an additional 24-hour culture, we added 20 μL of MTT to each well under dark conditions. The 96-well plate was then incubated in a 37 °C incubator for 4 hours. After removing the culture medium, we added 150 μL of DMSO to each well. After shaking on a room temperature rocker for 10 minutes, we measured the absorbance of each well at a wavelength of 490nm using a microplate reader. We calculated cell viability and selected three optimal concentrations of PMS for subsequent experiments.

To examine the protective effects of PMS on Min6 cells under HG and PA conditions, we conducted the following experimental protocol. Initially, Min6 cells were plated into 6-well plates at a density of $2 \times 10^5$ cells/well and cultured in normal 1640 complete medium containing 5.5mM glucose for 24 hours to facilitate cell attachment. Subsequently, the cells were allocated into five groups: (1). Normal Control group (NC): Cells maintained in normal 1640 medium. (2). HG and PA group (HG + PA): Cells treated with high glucose and high fat conditions containing 40mM glucose and 400 μM palmitic acid [22]. (3–5). PMSL, PMSM, and PMSH groups: Cells first exposed to HG and PA conditions for 48 hours and then treated with PMS at concentrations of 12.5, 25, 50 μM, respectively, for an additional 24 hours. Following the culture period, MTT assays were conducted to evaluate cell viability and determine the optimal concentration of PMS for subsequent experiments.

To explore the impact of PMS on ferroptosis and the xCT/GPX4 pathway in Min6 cells under HG and PA conditions, we conducted the following experiment. Initially, Min6 cells were plated into 6-well plates with $2 \times 10^5$ cells/well. Normal 1640 complete medium was used for culturing, which contained 5.5mM glucose for 24 hours to promote cell attachment. Subsequently, the cells were categorized into three groups: (1). NC: Cells maintained in normal 1640 medium. (2). HG + PA group: Cells treated with HG and PA conditions containing 40mM glucose and 400 μM palmitic acid. (3). PMS group: Cells

first exposed to HG and PA conditions for 48 hours. 50 μM of PMS was then used for treatment for an additional 24 hours. Following the culture period, cells and cell supernatants were gathered for subsequent assays.

To authenticate PMS protective effects on Min6 cells induced by HG and PA conditions via xCT/GPX4 pathway activation, we conducted the following experiments. Initially, Min6 cells were seeded into 6-well plates with $2 \times 10^5$ cells/well. Normal 1640 complete medium was used for culturing, which contained 5.5 mM glucose for 24 hours to facilitate cell attachment. Cells were then distributed into five groups: (1). NC: Cells maintained in normal 1640 medium. (2). HG + PA group: Cells treated with HG and PA conditions containing 40 mM glucose and 400 μM palmitic acid. (3). PMS group: Cells first exposed to HG and PA conditions for 48 hours. 50 μM of PMS was then used for treatment for an additional 24 hours. (4). Ferroptosis Inhibitor group (Fer-1): Cells first exposed to HG and PA conditions for 48 hours. 10 μM of Fer-1 was then used for treatment for an additional 24 hours [23]. (5). PMS + RSL-3 group: Cells first exposed to HG and PA conditions for 48 hours. 50 μM of PMS and 0.1 μM of RSL-3 was then used for treatment for an additional 24 hours [24]. Following the culture period, cells supernatants were harvested for assays to explore PMS effects on ferroptosis and the xCT/GPX4 pathway in Min6 cells under HG and PA conditions.

### 2.4 Index detection

**Biochemical index detection.** Frozen pancreatic tissues and collected cells were homogenized to prepare homogenates, and their levels were measured using malondialdehyde (MDA), 4-hydroxynonenal (4-HNE), reactive oxygen species (ROS), Iron, glutathione (GSH), and glutathione disulfide (GSSG) kits, respectively. Protein concentration was determined using a BCA kit to standardize the results of MDA, 4-HNE, ROS, Iron, GSH, and GSSG. Specific operational steps were conducted following the instructions provided in the kit manuals.

**RT-qPCR analysis.** Total RNA was extracted from 40 mg of frozen pancreatic tissue from each group. Reverse transcription and qPCR were performed under standard conditions. The expression levels of *Acsl4*, *ferritin light chain* (*Ftl*), *Transferrin* (*Trf*), *Steap3*, *solute carrier family 3 member 2* (*Slc3a2*), *solute carrier family 7A11* (*Slc7a11*), and *Gpx4* mRNA in pancreatic tissue from each group were calculated using the $2^{-\Delta\Delta CT}$ method. Primer sequences were designed and synthesized using the NCBI website as detailed in S1 Table.

**Western blot.** We extracted protein from the pancreatic tissues of mice in each group (with three samples per group) and from cells in each group. Then, the protein concentration was quantified using the BCA method. The expression of target proteins was conducted using western blot as described previously [25]. Image J was used to quantify the gray values of bands in western blot.

### 2.5 Statistics

We conducted statistical analysis using SPSS 22.0 software. All data were presented as mean ± SD. Group differences were assessed using one-way ANOVA, followed by the Tukey HSD test. A *P*-value of less than 0.05 was deemed statistically significant.

### 3. Result

#### Therapeutic effect of PMS on T2DM mice

Starting from the administration, we dynamically monitored the body weight and FBG changes of mice in each group on a weekly basis. In comparison to the CON group, the T2DM group exhibited progressive weight loss, elevated FBG levels, reduced glucose tolerance, increased HbA1c levels, and heightened homa-ir, indicating successful simulation of key characteristics of type 2 diabetes. However, following PMS treatment, these indicators showed improvement (Fig 1a-1f). Furthermore, H&E staining revealed that islets in the T2DM group, compared to the CON group, displayed disrupted tissue architecture characterized by atrophic deformation and structural disorganization, with ill-defined and irregular

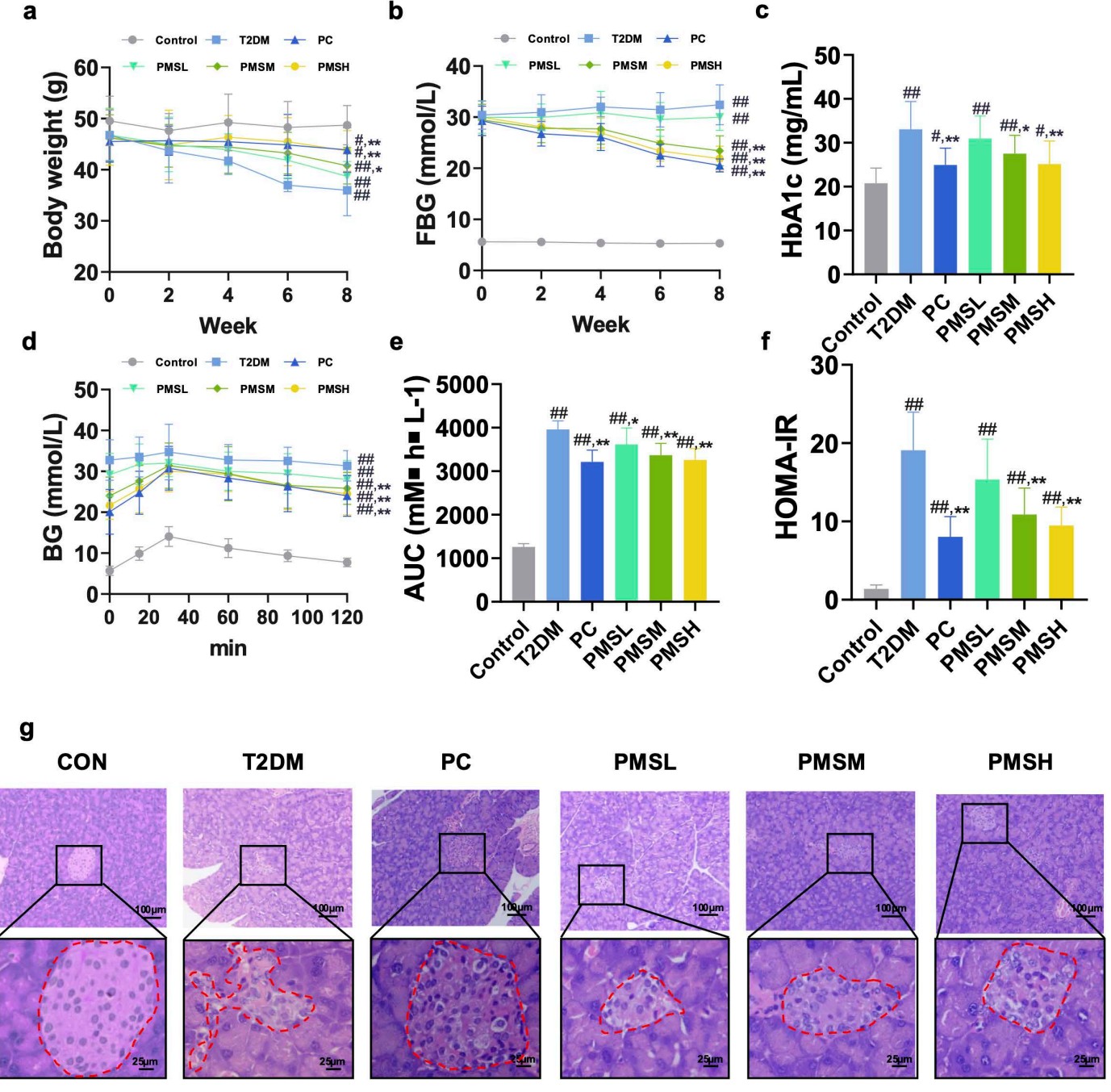

**Fig 1. PMS exhibited therapeutic effects on T2DM mice.** (a-f) PMS ameliorated the body weight loss (a) and decreased the levels of FBG (b), HbA1c (c), AUC of OGTT (d, e) and HOMA-IR (f). (g) H&E staining showed that PMS improved the pathological changes in pancreatic tissue of T2DM mice. Red dashed line: boundary of islet tissue. Data are presented as the mean ± SD, n = 10 per group. ##: $p < 0.01$ and #: $p < 0.05$ as compared to the control group; **: $p < 0.01$, and *: $p < 0.05$, as compared to the T2DM group.

contours. Conversely, the pathological morphology of pancreatic tissue in mice from all treatment groups demonstrated enhancement, with relatively preserved structures compared to the T2DM group (Fig 1g). These findings underscored a dose-dependent effect of PMS, substantiating its therapeutic efficacy in T2DM.

### Effect of PMS on Ferroptosis in the Pancreas of T2DM Mice

Iron overload initiates lipid peroxidation reactions, which constitute the primary hallmark of ferroptosis. Thus, we investigated the impact of PMS on ferroptosis by assessing lipid peroxidation and iron metabolism. TUNEL staining is widely known for detecting apoptosis by identifying DNA fragmentation, it can also be used to observe other types of cell death if DNA damage occurs, including ferroptosis under certain conditions. In our study, we utilized TUNEL staining to broadly detect cell death, not limited to apoptosis. Compared to the CON group, the T2DM group exhibited an elevated number of cell death. However, the PMS-treated groups exhibited a significant reduction in the number of cell death compared to the T2DM group (Fig 2a, 2b). Perl's Prussian blue staining demonstrated that the deposition of free iron in T2DM group was significantly increased compared to the CON group. However, the PMS-treated groups exhibited a significant reduction the deposition of free iron compared to the T2DM group (Fig 2c). Analysis of lipid peroxidation-related indicators demonstrated a significant elevation in MDA, 4-HNE, and ROS levels in the pancreatic tissue of T2DM mice compared to the CON group, indicative of extensive lipid peroxidation in the pancreas. Furthermore, the total iron level in the pancreatic tissue of T2DM mice was heightened, suggesting iron overload in the pancreas. Following PMS intervention, the levels of MDA, 4-HNE, and ROS were reduced in the pancreatic tissue of T2DM mice, accompanied by a decrease in the total iron level (Fig 2d-2g). ACSL4, FTL, TRF, and STEAP3 are pivotal proteins involved in iron metabolism, crucial for the ferroptosis process [26,27]. Therefore, we investigated these iron metabolism-related indicators. Our findings revealed a significant upregulation in the mRNA and protein expression levels of ACSL4, TRF, and STEAP3 in the pancreatic tissue of T2DM mice compared to the CON group, alongside a significant downregulation in the mRNA and protein expression of FTL, indicating iron accumulation and depletion in the pancreas. However, PMS intervention resulted in reduced mRNA and protein levels of ACSL4, TRF, and STEAP3 in the pancreatic tissue of T2DM mice, while upregulating the mRNA and protein levels of FTL (Fig 2k-2p).

### Effect of PMS on the xCT/GPX4 Pathway in the Pancreas of T2DM Mice

xCT/GPX4 pathways play a pivotal role in the antioxidant defense and iron metabolism of cells, and modulation of this pathway has emerged as a novel therapeutic approach for T2DM [12]. Hence, we investigated PMS effect on the xCT/GPX4 axis. Our findings revealed that GSH levels and GSH/GSSG ratios in T2DM pancreatic mice tissue were significantly lower compared to the CON group, accompanied by significantly higher GSSG levels, indicating GSH depletion in the pancreas (Fig 3a-3c). Moreover, xCT component proteins levels (SLC7A11 and SLC3A2), along with the lipid peroxide reductase GPX4, were significantly downregulated in T2DM mice pancreas compared to the CON group. This suggests suppression of the xCT/GPX4 axis T2DM mice pancreas, which depletes GSH and GPX4. However, treating with PMS significantly ameliorated all these symptoms, displaying a clear dose-dependent relationship. These results suggest that the mechanism underlying PMS inhibition of ferroptosis in T2DM mice pancreas may be associated with enhancement of the xCT/GPX4 axis (Fig 3d-3j).

### Protective effect of PMS on HG and PA -Induced injury in Min6 cells

In *vivo* experiments have demonstrated that PMS inhibits ferroptosis in the pancreatic tissue of T2DM mice while enhancing the xCT/GPX4 axis. However, it remains essential to explore whether PMS directly protects pancreatic beta cells. To address this, we conducted in *vitro* experiments using the HG and PA -induced Min6 cell model of pancreatic beta cells. MTT assay results indicated that PMS concentrations of 12.5, 25, and 50 μM had no significant impact on pancreatic beta cell viability, thus these concentrations were selected for further investigation (Fig 4a). Compared to the NC group, Min6

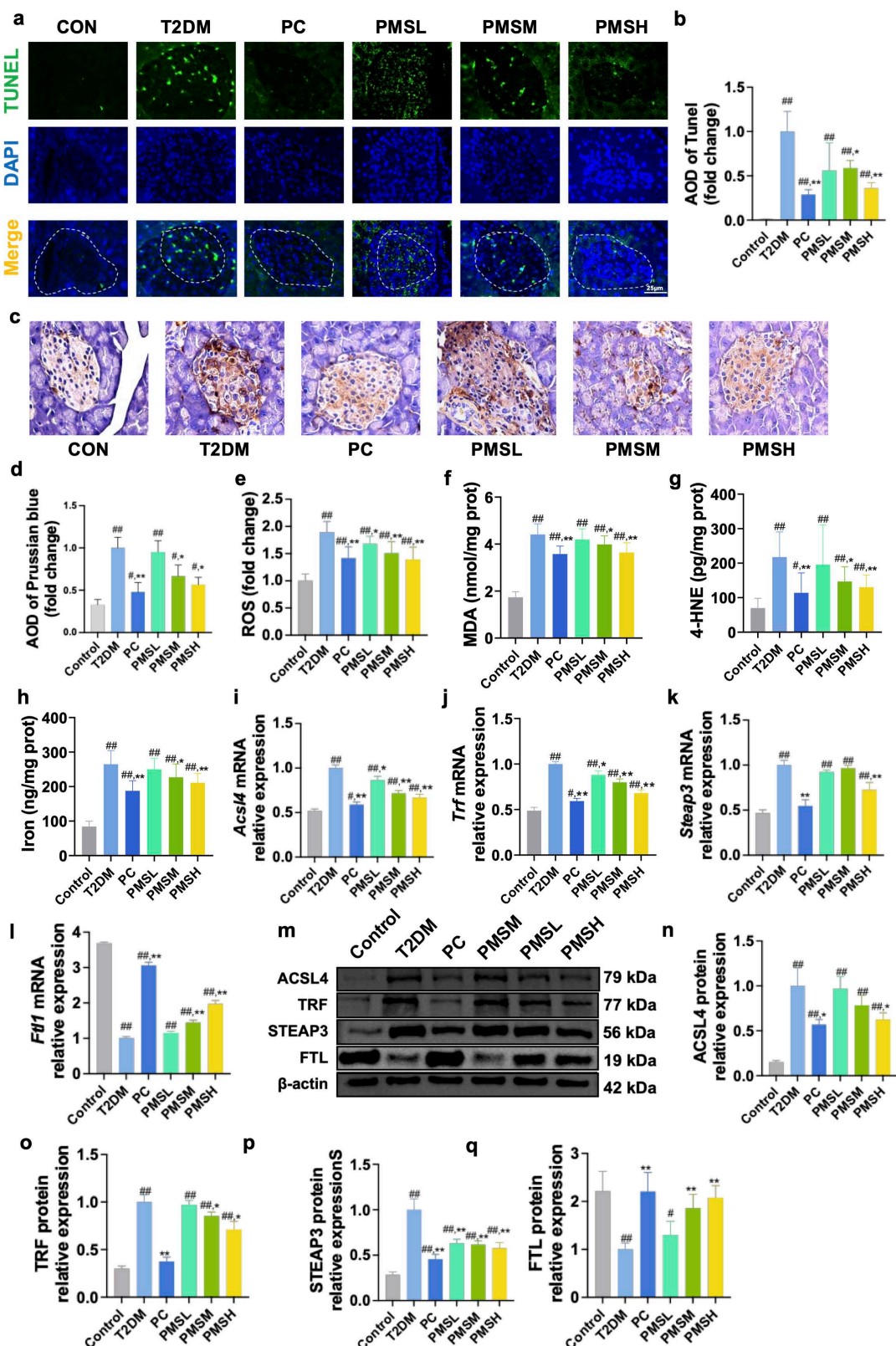

**Fig 2. PMS repressed ferroptosis in T2DM mice.** (a,b) TUNEL staining results demonstrated that the number of cell death in the pancreas of T2DM mice was significantly decreased following PMS intervention. The white dotted circle shows the location of the pancreatic islets. (c,d) Perl's Prussian

blue staining results demonstrated that the deposition of free iron in the pancreas of T2DM mice was significantly decreased following PMS intervention. (e-h) PMS intervention reduces levels of ROS (e), MDA (f), and 4-HNE (g), and decreased total Iron (h) level in pancreatic tissue of T2DM Mice. (i-l) RT-qPCR analysis showed that PMS reduced *Acsl4 Trf* (j), and *Steap3* (k) mRNA levels and increased *Ftl1* (l) mRNA levels. (m-q) Western blot results showed that PMSH intervention reduces the levels of ACSL4 (m, n), TRF (m, o), and STEAP3 (m, p) and increased the level of FTL (m, q). Data are presented as the mean±SD, n=3 for a-d; n=10 for e-h; n=3 for i-q. ##: $p < 0.01$ and #: $p < 0.05$ as compared to the control group; **: $p < 0.01$, and *: $p < 0.05$, as compared to the T2DM group.

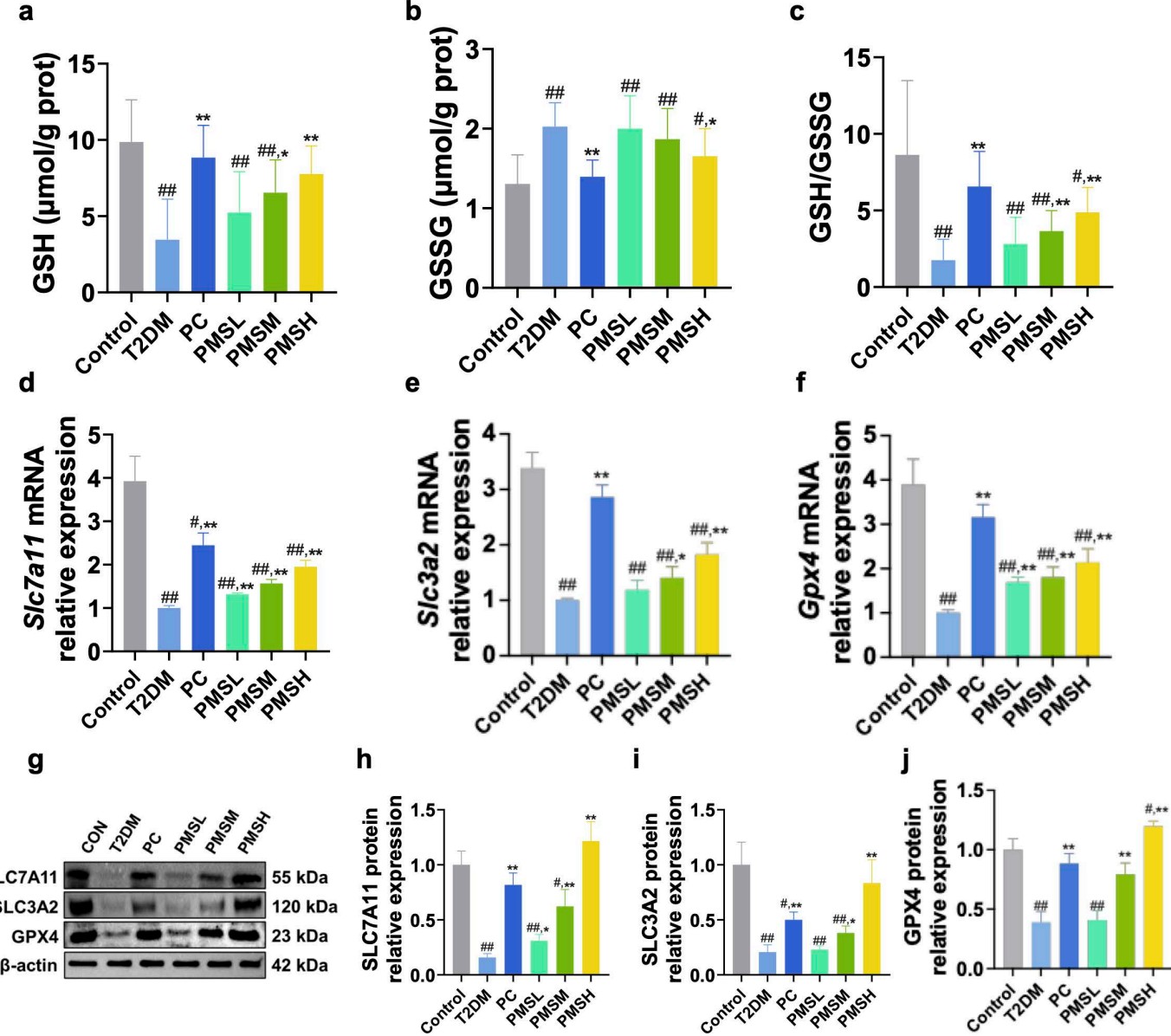

**Fig 3. PMS activated xCT/GPX4 pathway in T2DM mice.** (a-c) PMS increased GSH content (a), decreased GSSG level (b), and elevated the GSH/GSSG ratio (c) in pancreatic tissue. (d-f) RT-qPCR results showed that PMS up-regulated the mRNA expression of *Slc7a11* (d), *Slc3a2* (e), and *Gpx4* (f). (g-j) Western blot results showed that PMSH intervention increased the levels of SLC7A11 (g, h), SLC3A2 (g, i), and GPX4 (g, j). Data are presented as the mean±SD, n=10 for a-c; n=3 for d-j. ##: $p < 0.01$ and #: $p < 0.05$ as compared to the control group; **: $p < 0.01$, and *: $p < 0.05$, as compared to the T2DM group.

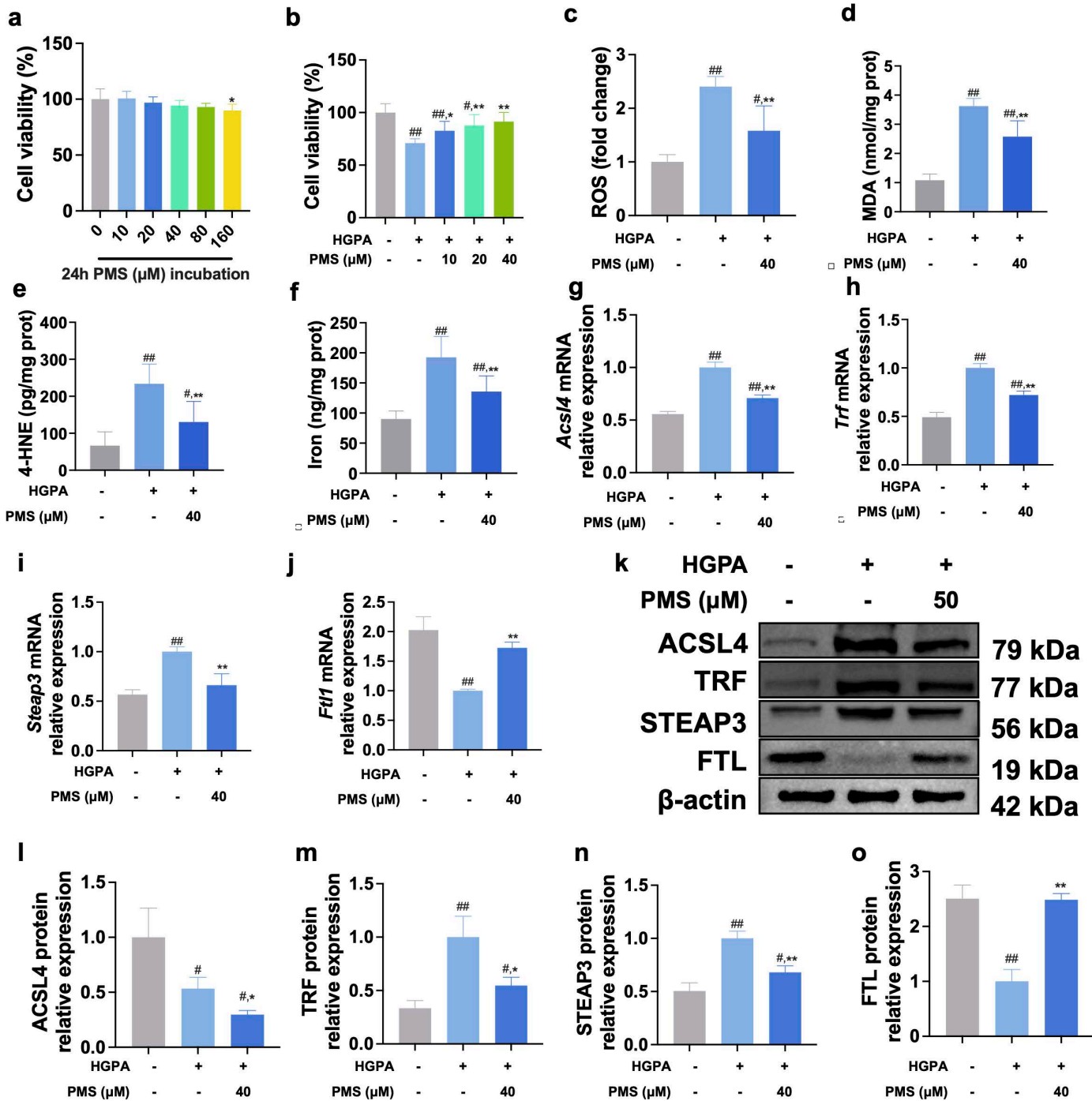

**Fig 4. PMS mitigated cell injury and inhibited ferroptosis in HG + PA treated Min6 cells.** (a, b) Cell viability was detected with an MTT assay, PMS (<80μM) did not affect Min6 pancreatic beta cell viability(a), and PMS restored viability in HG+PA-treated Min6 cells(b). (c-f) PMS reduced the levels of ROS (c), MDA (d), 4-HNE (e), and total Iron (f) in HG+PA treated Min6 cells. (g-j) RT-qPCR analysis showed that PMS downregulated the mRNA expression of *Acsl4* (g), *Trf* (h), *Steap3* (i) and up-regulated the mRNA expression of *Ftl1* (j). (k-o) Western blot results showed that PMSH intervention decreased the protein levels of ACSL4 (k, l), TRF (k, m), STEAP3 (k, n) and increased the protein level of FTL (k, o). Data are presented as the mean ± SD, n = 6 for a-f; n = 3 for g-o. ##: $p < 0.01$ and #: $p < 0.05$ as compared to the NC group; **: $p < 0.01$, and *: $p < 0.05$, as compared to the HG + PA group.

cells treated with HG and PA showed decreased viability, indicating HG and PA induced damage to pancreatic beta cells. Notably, PMS significantly enhanced the viability of Min6 cells after HG and PA induction, underscoring its protective role on pancreatic beta cells. Among the concentrations tested, PMS at 50 µM concentration exhibited the most pronounced effect and was therefore chosen for subsequent experiments (Fig 4b).

### Effects of PMS on Ferroptosis and xCT/GPX4 Pathway in Min6 Cells Induced by HG and PA

We investigated the direct effects of PMS intervention on ferroptosis and the xCT/GPX4 pathway in pancreatic beta cells. Results regarding lipid peroxidation-related indicators showed notable increases in MDA, 4-HNE, and ROS levels in the HG + PA group compared to the NC group, indicating significant lipid peroxidation within the cells. Furthermore, the total iron ion level in the HG + PA group was markedly higher than in the NC group, indicating substantial iron overload within the cells (Fig 4c-4f). RT-qPCR and Western blot analyses demonstrated significant upregulation of ACSL4, TRF, and Steap3 mRNA and protein expression in Min6 cells induced by HG and PA. PMS intervention reversed these effects (Fig 4g-4o).

Analysis of GSH and GSSG levels revealed a significant decrease in intracellular GSH levels and GSH/GSSG ratio in the HG + PA group compared to the NC group, along with a significant increase in GSSG levels, indicating pronounced GSH depletion within the cells (Fig 5a-5c). Finally, examination of genes and proteins related to the xCT/GPX4 pathway showed significant downregulation of SLC7A11, SLC3A2, and GPX4 levels in the HG + PA group compared to the NC group. These findings collectively suggest that Min6 cells induced by HG and PA undergo significant iron overload, leading to extensive lipid peroxidation and consequent depletion of GSH and GPX4, ultimately resulting in pronounced ferroptosis (Fig 5d-5j). PMS treatment notably improved all these symptoms, indicating its potential to alleviate ferroptosis in HG and PA -induced Min6 cells by enhancing the xCT/GPX4 axis.

### The impact of inhibiting GPX4 on the therapeutic effect of PMS in *Vitro*

While the previous results suggest that PMS can directly protect pancreatic beta cells by inhibiting ferroptosis and enhancing the xCT/GPX4 pathway, it remains uncertain whether PMS achieves this protective effect by activating the xCT/GPX4 axis and subsequently inhibiting ferroptosis. To further investigate this mechanism, we utilized the ferroptosis inhibitor Fer-1 as a positive control to compare its effects with PMS on ferroptosis in HG and PA-induced Min6 cells. Additionally, we employed a GPX4 inhibitor to assess the impact of inhibiting GPX4 on ferroptosis following PMS treatment. MTT assay results demonstrated that both Fer-1 and PMS exhibited protective effects on the viability of HG and PA -induced Min6 cells, with no significant difference in their therapeutic effects. However, RSL-3, the GPX4 inhibitor, nullified the therapeutic effect of PMS (Fig 6a). Moreover, in the analysis of lipid peroxidation and ferroptosis markers, Fer-1 showed similar effects to PMS, significantly reducing levels of ROS, MDA, and 4-HNE, and decreasing the accumulation of total iron ions (Fig 6b-6e). While significant downregulation of ACSL4, TRF, and Steap3 mRNA and protein expression, FTL mRNA and protein expression was significantly upregulated (Fig 6f-6n). Evaluation of xCT/GPX4 pathway-related markers revealed that both Fer-1 and PMS mitigated GSH depletion and upregulated the expression levels of SLC7A11, SLC3A2, and GPX4 mRNA and protein. Furthermore, intervention with RSL-3 abolished the therapeutic effects of PMS on lipid peroxidation, ferroptosis, and the xCT/GPX4 axis in HG and PA -induced Min6 cells (Fig 7a-7j). These findings suggest that PMS primarily reduces pancreatic beta cell damage by enhancing the xCT/GPX4 axis.

## 4. Discussion

T2DM represents a prevalent chronic metabolic condition characterized by impaired pancreatic beta cell function, inadequate insulin secretion, and insulin resistance [28]. Current research underscores the importance of enhancing insulin production in pancreatic beta cells and reducing insulin resistance as fundamental strategies for managing T2DM [29]. PMS is studied for protecting cells from stress and damage. The dosages administered in mouse models, ranging from 10

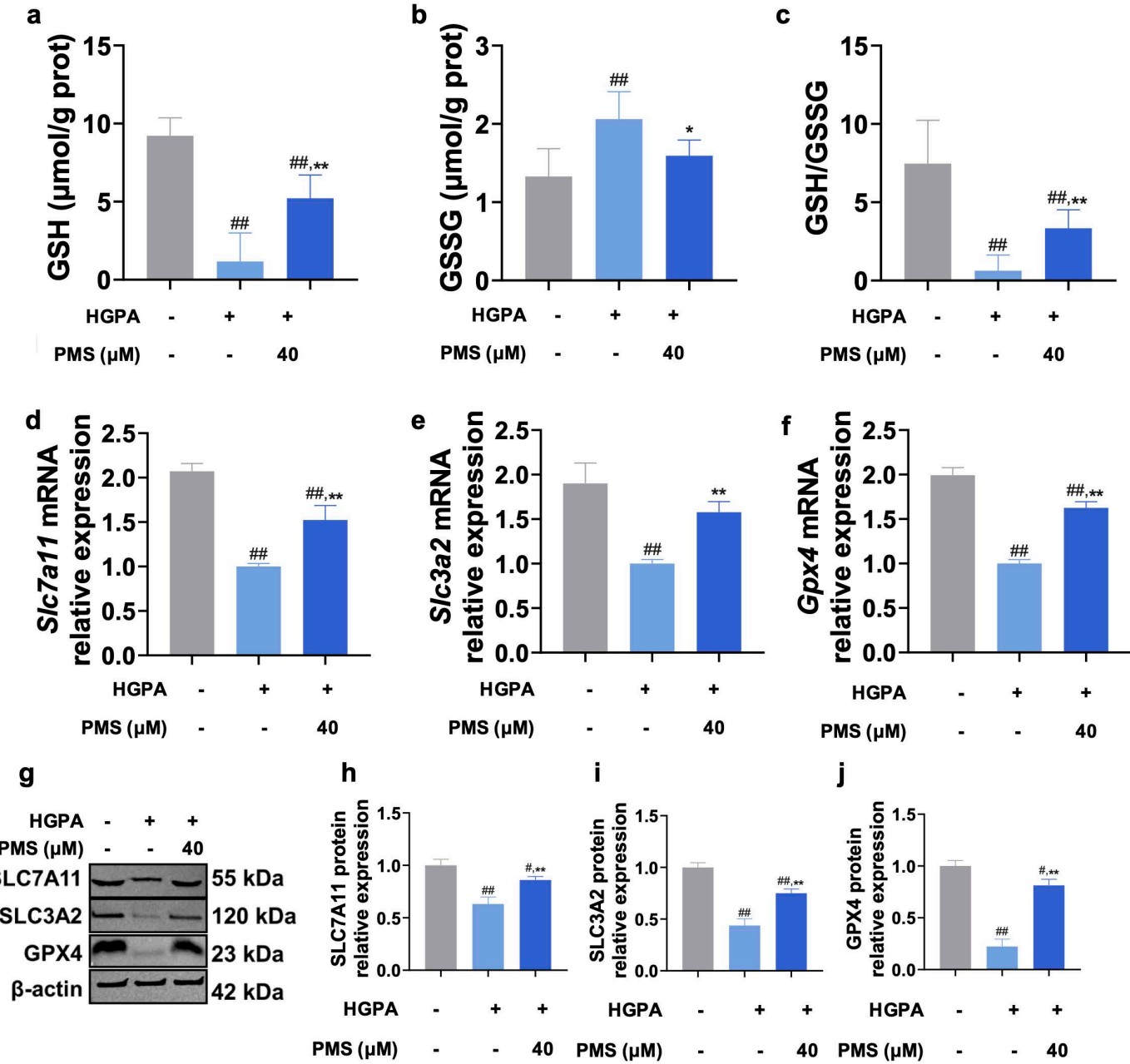

**Fig 5. PMS inhibited ferroptosis, improved GSH depletion and activated xCT/GPX4 pathway in Min6 cells after treated with HG + PA.** (a-c) PMS increased glutathione (GSH) content (a), decreased glutathione disulfide (GSSG) level (b), and elevated the GSH/GSSG ratio (c) in Min6 cells after treated with HG+PA. (d-f) RT-qPCR results showed that PMS upregulated the mRNA expression of *Slc7a11* (d), *Slc3a2* (e), and *Gpx4* (f). (g-j) Western blot results indicated that PMSH intervention increased the protein levels of SLC7A11 (g, h), SLC3A2 (g, i), and GPX4 (g, j). Data are presented as the mean ± SD, n = 6 for a-c; n = 3 for d-j. ##: $p < 0.01$ and #: $p < 0.05$ as compared to the NC group; **: $p < 0.01$, and *: $p < 0.05$, as compared to the HG+PA group.

to 100 mg/kg, have been applied to treat conditions such as caecal ligation and puncture, osteoarthritis, and LPS-induced PD [16–18]. Nevertheless, the therapeutic potential of PMS in T2DM and the mechanisms underlying its effects are not yet fully comprehended. Thus, this study aims to investigate, through both in *vivo* and in *vitro* experiments, whether PMS

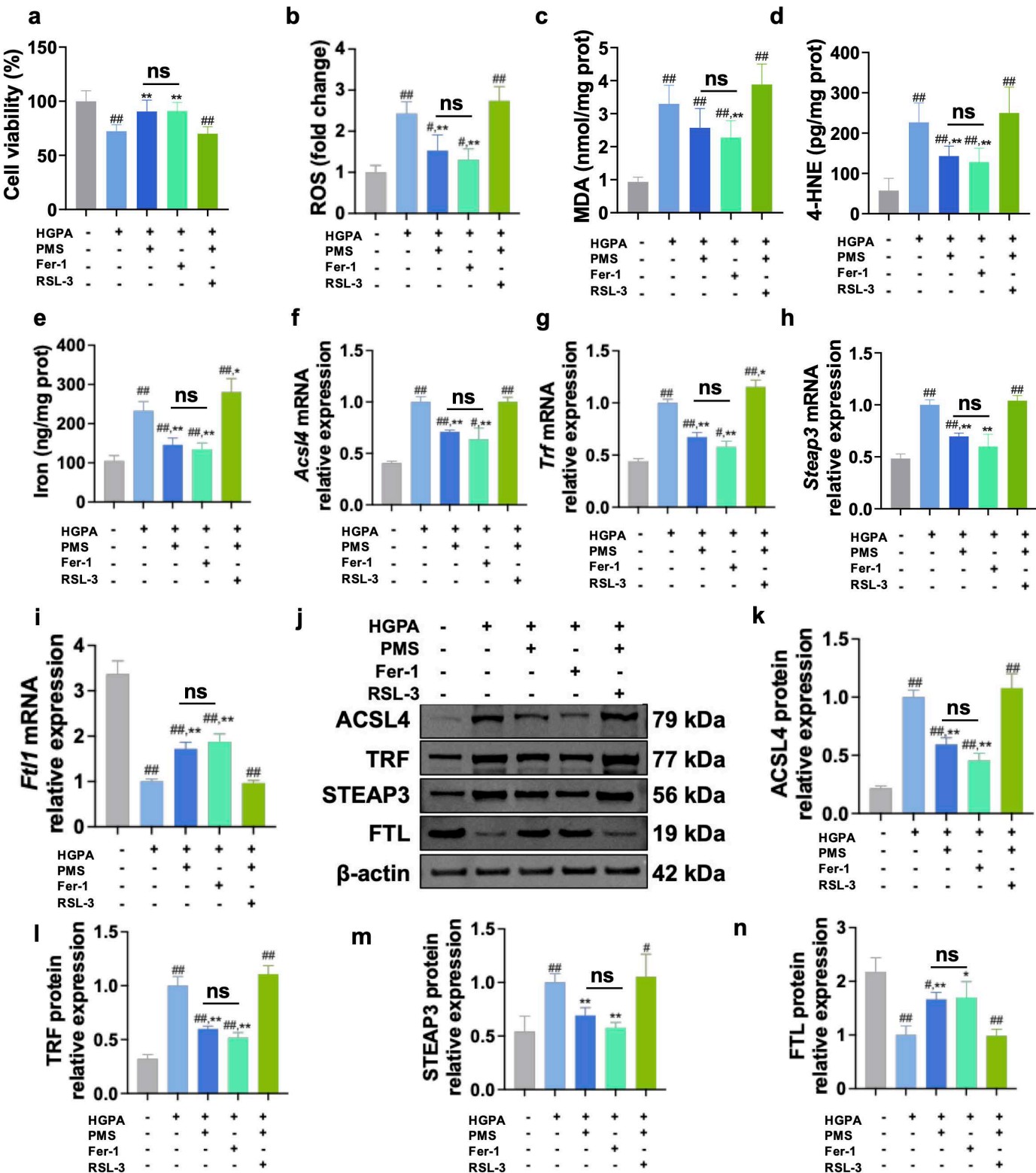

**Fig 6. GPX4 inhibition abolished the protective effects of PMS on HG + PA-induced Min6 cell damage.** (a) Cell viability was detected with a MTT assay, RSL-3 abolished the protective effects of PMS on HG + PA-induced Min6 cell injury. (b-e) RSL-3 abolished the effects of PMS on ROS (b), MDA (c), 4-HNE (d), and total Iron (e) in HG + PA -induced Min6 cells. (f-i) RT-qPCR analysis showed that RSL-3 abolished the effects of PMS on mRNA

expression of *Acsl4* (f), *Trf* (g), *Steap3* (h) and *Ftl1* (i). (j-n) Western blot results showed that RSL-3 abolished the effects of PMS on protein levels of ACSL4 (j, k), TRF (j, l) STEAP3 (j, m), and FTL (j, n). Data are presented as the mean ± SD, n = 6 for a-e; n = 3 for f-n. ##: $p < 0.01$ and #: $p < 0.05$ as compared to the NC group; **: $p < 0.01$, and *: $p < 0.05$, as compared to the HG + PA group; ns: no significant ($p > 0.05$).

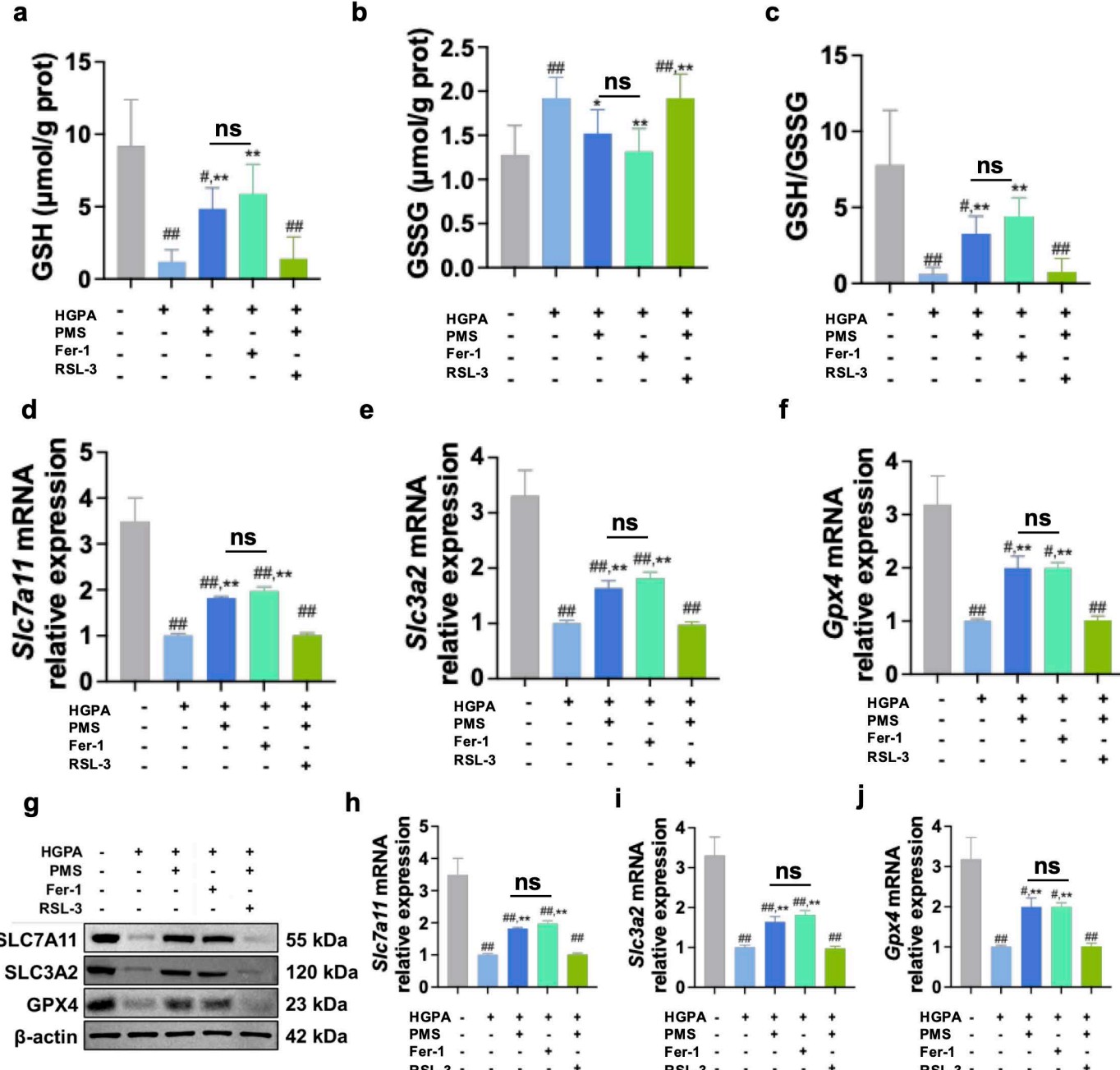

**Fig 7. GPX4 inhibition abolished modulatory effects of PMS on ferroptosis and xCT/GPX4 pathway.** (a-c) The effects of PMS on the levels of GSH (a) and GSSG (b), and on GSH/GSSG (c) ratio were abolished after RSL-3 treatment. (d-f) RT-qPCR results show that RSL-3 abolished the effects of PMS on the mRNA expression of *Slc7a11* (d), *Slc3a2* (e), and *Gpx4* (f). (g-j) Western blot results show that RSL-3 intervention abolished the effects of PMS on SLC7A11 (g,h), SLC3A2 (g,i), and GPX4 (g,j). Data are presented as the mean ± SD, n = 6 for a-c; n = 3 for d-j. ##: $p < 0.01$ and #: $p < 0.05$ as compared to the NC group; **: $p < 0.01$, and *: $p < 0.05$, as compared to the HG + PA group; ns: no significant ($p > 0.05$).

can effectively mitigate ferroptosis in pancreatic beta cells in T2DM by activating the xCT/GPX4 pathway, offering a potential novel therapeutic approach for T2DM treatment. In our in *vivo* experiments, we induced a T2DM mouse model using a high-fat diet combined with STZ injection and assessed the impact of PMS on pancreatic function. Our results revealed that STZ injection disrupted pancreatic beta cell function, resulting in elevated levels of FBG and HbA1c, increased blood insulin levels, and insulin resistance, indicating successful establishment of the T2DM model. Pathological analysis demonstrated structural damage to pancreatic tissue, characterized by irregular and blurred boundaries and vacuolar infiltration in the T2DM group. However, treatment with PMS led to improvements in pancreatic function and histopathological morphology of the islets across the various treatment groups, suggesting a protective effect of PMS on the pancreas.

Ferroptosis, a recently identified form of cell death, is closely linked with dysregulated iron metabolism. TUNEL staining was employed to detect potential DNA damage during ferroptosis. Results from pancreatic tissue TUNEL staining indicated that PMS can mitigate cell death in the pancreatic tissue of T2DM mice. Pancreatic beta cells are particularly vulnerable to ferroptosis, with notable iron accumulation occurring post-ferroptosis [30]. The primary mechanism underlying ferroptosis involves the disruption of redox homeostasis, characterized by heightened ROS levels and excessive accumulation of lipid peroxidation products such as MDA and 4-HNE [31]. Under high glucose and high fat conditions, MIN6 cells can mimic the microenvironment of pancreatic β-cells in T2DM patients, providing significant value for studying the pathogenesis and intervention strategies of T2DM. Although not a purely β-cell line, its mixed characteristics enable it to more comprehensively reflect changes in pancreatic endocrine function, offering unique advantages for T2DM research. Our study observed a significant reduction in elevated levels of iron, ROS, MDA, and 4-HNE in T2DM mice and HG and PA -induced Min6 cells following PMS treatment, indicating alleviation of iron overload and lipid peroxidation in the islets. Furthermore, PMS demonstrated the ability to modulate the expression levels of iron metabolism-related factors. ACSL4, TRF, STEAP3, and FTL are proteins intricately involved in iron metabolism and regulation. ACSL4, a crucial enzyme in fatty acid metabolism, plays a role in promoting phospholipid peroxidation and triggering ferroptosis, a pivotal characteristic of ferroptosis [26]. TRF, the primary iron transport protein, facilitates the delivery of $Fe^{3+}$ iron to target cells by binding to TRF receptors on the cell surface [32]. Subsequently, STEAP3 aids in the reduction of extracellular $Fe^{3+}$ to $Fe^{2+}$, facilitating intracellular iron uptake [33]. Intracellular iron is stored in ferritin, comprised of FTL, which prevents toxic reactions stemming from excessive iron accumulation. Breakdown of FTL during ferroptosis may lead to increased intracellular iron levels, further promoting ferroptosis occurrence [34].

Interestingly, although we utilized TUNEL staining to evaluate cell death in the pancreatic tissue of T2DM mice. However, TUNEL staining is a broader way to detect cell death, not limited to apoptosis. Meanwhile, pancreatic β-cell death is not caused by a single mechanism but is a result of multiple pathways, such as apoptosis, necrosis, and autophagy, which together lead to the loss of β-cell function and mass [35]. Apoptosis is primarily triggered by oxidative stress, endoplasmic reticulum (ER) stress, and pro-inflammatory cytokines. Under prolonged hyperglycemia, the accumulation of ROS and ER stress activate apoptotic pathways, leading to the activation of caspases and ultimately β-cell death [36]. Necroptosis, particularly under chronic inflammation and oxidative stress, plays an important role in β-cell loss. This process depends on the activation of RIPK1 and RIPK3, leading to cell membrane rupture and inflammation, which exacerbates β-cell damage [37]. Autophagy is a cellular self-degradation process that maintains cellular homeostasis by clearing damaged organelles. In β-cells, autophagy is essential for maintaining insulin secretion and cell survival. However, in T2DM, the inhibition of autophagy impairs β-cells' ability to clear oxidative damage and lipid overload, contributing to insulin secretion failure [38]. Despite extensive studies on these mechanisms, ferroptosis, a newly identified form of cell death, has increasingly gained attention. In addition, the interaction between iron death and other modes of cell death, such as apoptosis, necrosis and autophagy, is an important direction for future research.

Further studies have shown that PMS significantly ameliorates GSH depletion in T2DM mice and HG and PA -induced Min6 cells, along with enhancing SLC3A2, and GPX4 protein expression levels. These findings suggest that PMS may mitigate ferroptosis by bolstering the xCT/GPX4 axis. The xCT/GPX4 pathway stands as a pivotal regulator of lipid

peroxidation, crucial for thwarting ferroptosis [39]. This pathway encompasses the cystine/glutamate antiporter (system Xc-), on the cell membrane, a sodium-independent amino acid transport system, consisting of the light chain subunit SLC7A11 (xCT) and the heavy chain subunit SLC3A2. Notably, SLC7A11 predominantly governs its transport activity [40]. System Xc- facilitates the efflux of glutamate out of the cell while importing cystine, subsequently converted into cysteine, a precursor for glutathione (GSH) synthesis [41]. However, the expression levels of SLC7A11 may be dynamically changing during the initiation and progression of ferroptosis. During the early stages of ferroptosis or moderate oxidative stress, SLC7A11 may be upregulated as a compensatory mechanism to enhance cystine uptake and glutathione synthesis [7,42]. However, in conditions of prolonged or severe metabolic stress, such as glucotoxicity and lipotoxicity in β-cells, the sustained depletion of GSH or other metabolic constraints might overwhelm this compensatory mechanism. This could result in an inability to maintain high SLC7A11 expression, leading to its downregulation. This dynamic relationship could explain the observed differences between our study and previous findings under acute ferroptosis-inducing conditions. GPX4, as the linchpin of the antioxidant functional axis, utilizes GSH as a substrate to metabolize accumulated lipid peroxidation products into non-toxic lipid alcohols, thus rectifying lipid peroxidation damage and counteracting ferroptosis [43]. Furthermore, PMS treatment resulted in an increase of GSH/GSSG ratio with higher GSH levels and lower GSSG level. During ferroptosis, the reduction in GSH synthesis and the decline in GPX4 activity lead to an inability to effectively reduce lipid peroxides to their corresponding alcohols, resulting in the accumulation of lipid peroxides. Concurrently, when cells are subjected to oxidative stress, GSH is oxidized to GSSG to eliminate ROS, leading to a relative increase in GSSG levels and a decrease in the GSH/GSSG ratio [44]. However, it is important to note that GSSG levels are not always simply increased or unchanged; the levels of GSH and GSSG may dynamically fluctuate. A decrease in the GSH/GSSG ratio is commonly indicative of the occurrence of lipid peroxidation and ferroptosis. Thus, ensuring optimal xCT function is imperative for maintaining adequate substrates for GSH synthesis. With ample GSH and active GPX4, lipid peroxides can be efficiently cleared, thereby preventing the occurrence of ferroptosis.

Additionally, to investigate whether PMS mitigates T2DM by inhibiting ferroptosis through the Xc(-)/GPX4 functional axis, this experiment incorporated a ferroptosis inhibitor Fer-1 group and a GPX4 inhibitor RSL-3 group for comparative analysis. RSL-3 serves as a targeted inhibitor of GPX4, binding to its active site of selenocysteine and directly deactivating GPX4, thereby disrupting the antioxidant functional axis [45]. Results from the study indicated that Fer-1 notably attenuated damage to HG and PA -induced Min6 cells, evidenced by enhanced cell viability and improved indicators related to ferroptosis and lipid peroxidation. Furthermore, there was no significant disparity observed between the therapeutic effects of Fer-1 and PMS. Conversely, RSL-3 impeded GPX4 activity in Min6 cells, precipitating ferroptosis. Notably, the results revealed that RSL-3 nullified all therapeutic effects of PMS. These findings substantiate that PMS primarily diminishes ferroptosis and treats T2DM by enhancing the xCT/GPX4 axis.

In this investigation, metformin, Fer-1, and PMS all demonstrate therapeutic advantages for T2DM. As the first-line medication for T2DM, metformin exerts its efficacy through insulin sensitivity enhancement, though prolonged administration may induce gastrointestinal complications (e.g., diarrhea) and vitamin B12 deficiency risks [3]. Fer-1, a specific ferroptosis inhibitor [46], effectively suppresses lipid peroxidation. However, its clinical safety profile remains unverified. In contrast, PMS – a phenylethanoid glycoside derived from the traditional medicinal herb Plantago asiatica L. – exhibits superior biocompatibility due to its natural origin. Chronic toxicity studies revealed no hepatorenal dysfunction at doses up to 200 mg/kg [9], with histopathological analyses confirming absence of tissue abnormalities [16], collectively indicating minimal toxicity in animal models. Furthermore, PMS demonstrates pleiotropic mechanisms including ferroptosis inhibition, anti-inflammatory activity, and antioxidant capacity [11], which may synergistically preserve β-cell integrity. Although Fer-1 and metformin show superior efficacy under specific experimental conditions, PMS' natural derivation and multimodal mechanisms confer distinctive advantages for long-term therapeutic regimens.

A recent study and our work both investigate PMS' therapeutic potential in T2DM [47]. Our research elucidates PMS-mediated mitigation of pancreatic β-cell damage through ferroptosis inhibition, particularly via antioxidant effects and

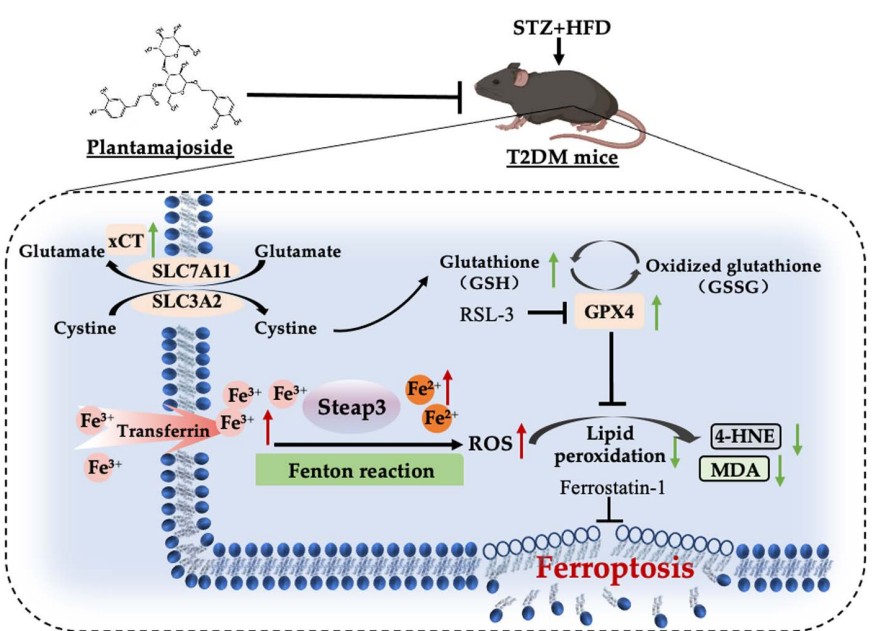

**Fig 8. PMS shows promise as a potential therapeutic agent for T2DM, potentially acting by inhibiting ferroptosis of pancreatic β-cells through activation of the xCT/GPX4 pathway.**

xCT/GPX4 pathway activation. However, Wang et al. employed transcriptomic profiling to delineate PMS' β-cell protective mechanisms through endoplasmic reticulum stress (ERS) and apoptosis modulation [47]. Notably, mechanistic crosstalk may exist between these pathways: ferroptosis frequently coincides with oxidative stress and lipid peroxidation, while ERS potentially exacerbates oxidative damage via the unfolded protein response. Future investigations should prioritize deciphering these inter-pathway interactions to comprehensively unravel PMS' β-cell preservation mechanisms, thereby strengthening the preclinical foundation for its clinical translation.

Furthermore, we plan to conduct future experiments to silence GPX4 and other upstream components of the pathway, utilizing techniques such as CRISPR/Cas9 or siRNA. Additionally, we will employ methods like Cellular Thermal Shift Assay (CETSA) to explore potential direct interactions of PMS with target proteins within the xCT/GPX4 pathway. These approaches will help us to elucidate the precise mechanism by which PMS inhibits ferroptosis and its target within the pathway.

## 5. Conclusion

PMS exhibited the capacity to diminish damage to pancreatic islet β-cells induced by T2DM, both in *vivo* and in *vitro*. This favorable outcome may stem from the alleviation of lipid peroxidation and reduction of ferroptosis. Moreover, this regulatory mechanism was accomplished through the enhancement of the xCT/GPX4 axis (Fig 8).

## Supporting information

**S1 Data. The details of materials, and reagents.**
(PDF)

**S1 Table. Primer sequence.**
(DOCX)

**S2 Data. Original Image WB.**
(PDF)

## Author contributions

**Conceptualization:** Shuquan Lv, Weibo Wen.

**Data curation:** Renlin Li, Xuan Guo, Jingrui Kang, Shuquan Lv, Zhongyong Zhang.

**Formal analysis:** Huajun Li, Xiaoyun Wang.

**Funding acquisition:** Hongmin Zhao.

**Investigation:** Hongmin Zhao, Renlin Li, Xuan Guo, Jingrui Kang, Yuansong Wang, Huantian Cui, Shuquan Lv, Zhongyong Zhang.

**Validation:** Renlin Li, Xuan Guo, Jingrui Kang, Yuansong Wang, Huantian Cui.

**Visualization:** Huajun Li, Xiaoyun Wang.

**Writing – original draft:** Renlin Li, Huajun Li, Xiaoyun Wang, Shuquan Lv, Weibo Wen, Zhongyong Zhang.

**Writing – review & editing:** Hongmin Zhao.

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
