## [Decision Letter · Decision Letter 0]

Apr 19 2025

PONE-D-25-01394Mechanism of Plantamajoside in inhibiting ferroptosis of pancreatic β cells and treatment of T2DM via activation of the xCT/GPX4 pathwayPLOS ONE

Dear Dr. Wen,

Thank you for submitting your manuscript to PLOS ONE. After careful consideration, we feel that it has merit but does not fully meet PLOS ONE’s publication criteria as it currently stands. Therefore, we invite you to submit a revised version of the manuscript that addresses the points raised during the review process.

We look forward to receiving your revised manuscript.

Kind regards,

Kai Huang

Academic Editor

PLOS ONE

2. To comply with PLOS ONE submissions requirements, in your Methods section, please provide additional information regarding the experiments involving animals and ensure you have included details on (1) methods of sacrifice, and (2) efforts to alleviate suffering.

Reviewers' comments:

Reviewer's Responses to Questions

**Comments to the Author**

1. Is the manuscript technically sound, and do the data support the conclusions?

Reviewer #1: Yes

Reviewer #2: Yes

2. Has the statistical analysis been performed appropriately and rigorously? 

Reviewer #1: Yes

Reviewer #2: Yes

3. Have the authors made all data underlying the findings in their manuscript fully available?

Reviewer #1: Yes

Reviewer #2: Yes

4. Is the manuscript presented in an intelligible fashion and written in standard English?

Reviewer #1: Yes

Reviewer #2: Yes

5. Review Comments to the Author

Reviewer #1: Overall, this study offers important insights into the role of Plantamajoside in ferroptosis and its potential application in treating Type 2 Diabetes Mellitus (T2DM), but it requires several revisions to improve readability. Considering that this journal is multidisciplinary and has readers from diverse backgrounds, the manuscript needs substantial improvements in the presentation of figures and the writing.

Issue 1: It seems that this paper was originally formatted according to the guidelines of journals other than PLOS ONE. Please reorganize the Abstract into one paragraph, according to the PLOS ONE template. In addition, please reorganize the Methods section.

Issue 2: Before using an abbreviation, the authors should ensure that they provide the full name at its first appearance. For example, “STZ,” “HbA1c,” “HOMA-IR,” “H&E,” and “TUNEL” in Method section 2.2, and “AOD” in Fig 2d. In the Abbreviations section, “Transferrin, TRF” should be written in order as “TRF, Transferrin.”

Issue 3: The Introduction section should state the rationale between ferroptosis and the XCT/GPX pathway."

Issue 4: In Figure 1g, readers may find it difficult to understand the conveyed messages from the current version, as the figures seem to show no obvious differences except in color at first glance. For better presentation, please label the different regions or lesions in each figure. You could add arrows, lines, or circles, for instance, to indicate “disrupted pancreatic tissue structure, irregular and blurred boundaries,” and “vacuolar infiltration.”

Issue 5: What is the n per group in Figures 2-7? Only Figure 1 states that n=10 per group.

Issue 6: Please explain or clarify the discrepancies in protein molecular weights. Mouse ACSL4 is approximately 79 kDa, but it shows ~19 kDa in Figure 2m. Mouse SLC3A2 should be approximately 55 kDa, but it shows about 120 kDa in Figure 3g.

Issue 7: The figure legends do not match the sub-figures. For example, Figures 3b-c and Figures 3d-e.

Please also check the following statements:

Figure 2, “Western blot results showed that PMS intervention reduces the levels of ACSL4 (m, n), TRF (m, o), and Steap3 (m, p) and increased the level of FTL (m, q).”

Figure 3, “Western blot results showed that PMS intervention increased the levels of SLC3A2 (g, h), SLC7A11 (g, i), and GPX4 (g, j).”

Figure 4, “Western blot results showed that PMS intervention decreased the protein levels of ACSL4 (k, l), TRF (k, m), Steap3 (k, n) and increased the protein level of FTL (k, o)”

Figure 5, “Western blot results indicated that PMS intervention increased the protein levels of SLC3A2 (g, h), SLC7A11 (g, i), and GPX4 (g, j).”

Figure 6, “RT-qPCR analysis showed that RSL3 abolished the effects of PMS on mRNA expression of Acsl4 (f), Trf (j), Steap3 (h) and Ftl (i).”

Figure 7, “Western blot results show that RSL3 intervention abolished the effects of PMS on SLC3A2 (g,h), SLC7A11 (g,i), and GPX4 (g,j).”

Issue 8: Figure 4a shows that cell viability decreases as PMS concentration increases, but the authors state that PMS did not exhibit cytotoxicity. Please clarify this discrepancy.

Issue 9: Several western bands look weird. Please provides all the raw images. The authors should upload all the raw images of the western blots as supplementary files or figures.

Issue 10: The authors should ensure that all the raw data are provided as supplementary files or figures. Specifically, please provide Excel files (or tables) containing the raw data related to all the test groups in Figures 1-7.

Issue 11: why not also use Fer-1 as a positive control in figure 1-5?

Reviewer #2: In this paper, the authors demonstrate that Plantamajoside (PMS) is able to reduce ferroptosis in pancreatic β cells from type II diabetes in vivo models, with similar effects than Metformin. In a cell model, they confirmed the effect of PMS and showed that these are modulated through the xCT/GPX4 pathway. This paper will be of interest for the type II diabetes research community, with potential for developing PMS or PMS-derived novel therapies. Here are some comments:

1) It would be nice to add some discussion comparing potential advantage/disadvantage of Metformin/Fer-1/PMS as treatment for type II diabetes as these two compounds seem to outperform PMS in the study.

2) A very similar study was very recently published, it would be nice to discuss their results in comparison to this study: 10.4239/wjd.v16.i2.99053

3) The quality of the figures is very low, please ensure they are clear for readability.

6. PLOS authors have the option to publish the peer review history of their article (what does this mean? ). If published, this will include your full peer review and any attached files.

**Do you want your identity to be public for this peer review?** For information about this choice, including consent withdrawal, please see our Privacy Policy .

Reviewer #1: No

Reviewer #2: No

---

## [Author Response · Author response to Decision Letter 1]

22 Apr 2025

Dear editor and reviewers,

Thanks for your comments, we have carefully edited our manuscript. The edited parts have been marked in red in our revised manuscript. The point-by-point responses of your comments are shown as follows:

Reviewer #1:

Overall, this study offers important insights into the role of Plantamajoside in ferroptosis and its potential application in treating Type 2 Diabetes Mellitus (T2DM), but it requires several revisions to improve readability. Considering that this journal is multidisciplinary and has readers from diverse backgrounds, the manuscript needs substantial improvements in the presentation of figures and the writing.

Issue 1: It seems that this paper was originally formatted according to the guidelines of journals other than PLOS ONE. Please reorganize the Abstract into one paragraph, according to the PLOS ONE template. In addition, please reorganize the Methods section.

Response: We have reorganized the abstract according to the PLOS ONE template and also reorganized the Methods section.

Issue 2: Before using an abbreviation, the authors should ensure that they provide the full name at its first appearance. For example, “STZ,” “HbA1c,” “HOMA-IR,” “H&E,” and “TUNEL” in Method section 2.2, and “AOD” in Fig 2d. In the Abbreviations section, “Transferrin, TRF” should be written in order as “TRF, Transferrin.”

Response: Thank you for your detailed review. We have checked the entire text and added the full name of abbreviations and also modified the "Abbreviations" section.

Issue 3: The Introduction section should state the rationale between ferroptosis and the XCT/GPX pathway."

Response: We have revised the Introduction section to explicitly clarify the mechanistic link between ferroptosis and the XCT/GPX pathway.

Issue 4: In Figure 1g, readers may find it difficult to understand the conveyed messages from the current version, as the figures seem to show no obvious differences except in color at first glance. For better presentation, please label the different regions or lesions in each figure. You could add arrows, lines, or circles, for instance, to indicate “disrupted pancreatic tissue structure, irregular and blurred boundaries,” and “vacuolar infiltration.”

Response: We have added relevant annotations in Figure 1g.

Issue 5: What is the n per group in Figures 2-7? Only Figure 1 states that n=10 per group.

Response: We have increased the n for each group in Figures 2-7.

Issue 6: Please explain or clarify the discrepancies in protein molecular weights. Mouse ACSL4 is approximately 79 kDa, but it shows ~19 kDa in Figure 2m. Mouse SLC3A2 should be approximately 55 kDa, but it shows about 120 kDa in Figure 3g.

Response: We sincerely appreciate your careful review and insightful comments. We have repeated the experiment for ACSL4, and the corrected results now show ACSL4 at its expected molecular weight of ~79 kDa. The initial discrepancy (19 kDa) was likely due to antibody non-specific binding or technical issues, and the revised data is included in the manuscript. We apologize for the initial error.

Regarding SLC3A2 (CD98hc), it is a membrane protein that undergoes glycosylation and forms a disulfide-bonded heterodimer with a non-glycosylated light chain, resulting in a complex of ~120-130 kDa (PMID: 14770309). The observed molecular weight (~120 kDa) in our experiments aligns with this heterodimeric form. Importantly, metabolic stress conditions (e.g., high glucose and palmitic acid) can enhance this structure, which is central to our study of ferroptosis in pancreatic β-cells. We are currently investigating the role of metabolic stress in modulating SLC3A2 molecular weight and will include these findings in future work.

Issue 7: The figure legends do not match the sub-figures. For example, Figures 3b-c and Figures 3d-e.

Response: We revised the legends and thoroughly checked the full text and sub-figure captions.

Please also check the following statements:

Figure 2, “Western blot results showed that PMS intervention reduces the levels of ACSL4 (m, n), TRF (m, o), and Steap3 (m, p) and increased the level of FTL (m, q).”

Response: We revised the legends in Figure 2.

Figure 3, “Western blot results showed that PMS intervention increased the levels of SLC3A2 (g, h), SLC7A11 (g, i), and GPX4 (g, j).”

Response: We revised the legends in Figure 3.

Figure 4, “Western blot results showed that PMS intervention decreased the protein levels of ACSL4 (k, l), TRF (k, m), Steap3 (k, n) and increased the protein level of FTL (k, o)”

Response: We revised the legends in Figure 4.

Figure 5, “Western blot results indicated that PMS intervention increased the protein levels of SLC3A2 (g, h), SLC7A11 (g, i), and GPX4 (g, j).”

Response: We revised the legends in Figure 5.

Figure 6, “RT-qPCR analysis showed that RSL3 abolished the effects of PMS on mRNA expression of Acsl4 (f), Trf (j), Steap3 (h) and Ftl (i).”

Response: We revised the legends in Figure 6.

Figure 7, “Western blot results show that RSL3 intervention abolished the effects of PMS on SLC3A2 (g,h), SLC7A11 (g,i), and GPX4 (g,j).”

Response: We revised the legends in Figure 7.

Issue 8: Figure 4a shows that cell viability decreases as PMS concentration increases, but the authors state that PMS did not exhibit cytotoxicity. Please clarify this discrepancy.

Response: We have revised the original statement 'PMS did not exhibit cytotoxicity' in the figure legend to a more rigorous description version: 'PMS (<80μM) did not affect Min6 pancreatic beta cell viability.'"

Issue 9: Several western bands look weird. Please provides all the raw images. The authors should upload all the raw images of the western blots as supplementary files or figures.

Response: We have upload all the raw images of the western blots as supplementary files.

Issue 10: The authors should ensure that all the raw data are provided as supplementary files or figures. Specifically, please provide Excel files (or tables) containing the raw data related to all the test groups in Figures 1-7.

Response: We have upload all the Excel files (or tables) containing the raw data related to all the test groups in Figures 1-7.

Issue 11: why not also use Fer-1 as a positive control in figure 1-5?

Response:In the in vivo study (Figs. 1-3), our primary focus was to evaluate the therapeutic efficacy of PMS on T2DM using a clinically relevant positive control. Metformin (MET), a first-line antidiabetic drug, was selected because it directly mirrors the intended therapeutic application of PMS in T2DM management (PMID: 34385345). The primary purpose of using Fer-1 was to clarify whether PMS exerts its therapeutic effects on T2DM through ferroptosis inhibition. In the in vitro experiments, Fer-1 was employed as a positive control (Figs. 6-7). The results demonstrated no significant differences between PMS and Fer-1 in improving HG+PA-induced Min6 cell damage, enhancing antioxidant capacity, or suppressing ferroptosis..

Reviewer #2:

In this paper, the authors demonstrate that Plantamajoside (PMS) is able to reduce ferroptosis in pancreatic β cells from type II diabetes in vivo models, with similar effects than Metformin. In a cell model, they confirmed the effect of PMS and showed that these are modulated through the xCT/GPX4 pathway. This paper will be of interest for the type II diabetes research community, with potential for developing PMS or PMS-derived novel therapies. Here are some comments:

1) It would be nice to add some discussion comparing potential advantage/disadvantage of Metformin/Fer-1/PMS as treatment for type II diabetes as these two compounds seem to outperform PMS in the study.

Response:We sincerely appreciate this valuable comment. Previous studies have established that while metformin remains the first-line therapeutic agent for T2DM through insulin sensitivity improvement (PMID: 38138975), its long-term use may be accompanied by gastrointestinal adverse effects (e.g., diarrhea) and vitamin B12 deficiency risks. Ferrostatin-1 (Fer-1), as a specific ferroptosis inhibitor (PMID: 31574461), effectively blocks lipid peroxidation. However, its clinical safety profile remains unverified. By comparison, PMS – a phenylethanoid glycoside isolated from the traditional medicinal herb Plantago asiatica L. – demonstrates superior biocompatibility owing to its natural origin. Chronic administration studies revealed no hepatorenal dysfunction at doses up to 200 mg/kg (PMID: 24316425), with histopathological examinations confirming absence of tissue abnormalities (PMID: 37288729), collectively indicating negligible toxicity in animal models. Furthermore, PMS exhibits pleiotropic mechanisms encompassing ferroptosis suppression, anti-inflammatory activity, and antioxidant capacity (PMID: 32605460), which may synergistically protect β-cells. Although both Fer-1 and metformin showed stronger effects under specific experimental conditions, PMS' natural derivation and multimodal mechanisms confer distinctive advantages for long-term therapeutic applications. Subsequent investigations should prioritize validation of its chronic efficacy and pharmacokinetic properties. We deeply value this constructive feedback, which significantly informs our research team's strategic direction.

2) A very similar study was very recently published, it would be nice to discuss their results in comparison to this study: 10.4239/wjd.v16.i2.99053

Response:A recent study (PMID: 39959264) and our investigation both explored the therapeutic effects of PMS on T2DM. This study elucidated the mechanism of PMS in alleviating pancreatic β-cell damage through ferroptosis inhibition, demonstrating its potent antioxidant effects via activation of the xCT/GPX4 pathway. In contrast, the research by Wang et al. employed transcriptomic analysis to reveal PMS-mediated β-cell protection from endoplasmic reticulum stress (ERS) and apoptosis. Furthermore, the mechanisms identified in these two studies may exhibit potential crosstalk: ferroptosis frequently coexists with oxidative stress and lipid peroxidation, while ERS may exacerbate oxidative damage through the unfolded protein response. Future investigations should prioritize exploring these mechanistic interactions to comprehensively delineate PMS' β-cell protective mechanisms, thereby providing robust theoretical support for its clinical translation.

3) The quality of the figures is very low, please ensure they are clear for readability.

Response:We have reuploaded the high-definition figures(300dpi).

If you have more comments on our manuscript, please do not hesitate to contact us.

Best regards,

Huantian Cui

---

## [Decision Letter · Decision Letter 1]

Jun 20 2025

PONE-D-25-01394R1Mechanism of Plantamajoside in inhibiting ferroptosis of pancreatic β cells and treatment of T2DM via activation of the xCT/GPX4 pathwayPLOS ONE

Dear Dr. Wen,

Thank you for submitting your manuscript to PLOS ONE. After careful consideration, we feel that it has merit but does not fully meet PLOS ONE’s publication criteria as it currently stands. Therefore, we invite you to submit a revised version of the manuscript that addresses the points raised during the review process.

We look forward to receiving your revised manuscript.

Kind regards,

Kai Huang

Academic Editor

PLOS ONE

Journal Requirements:

Reviewers' comments:

Reviewer's Responses to Questions

**Comments to the Author**

1. If the authors have adequately addressed your comments raised in a previous round of review and you feel that this manuscript is now acceptable for publication, you may indicate that here to bypass the “Comments to the Author” section, enter your conflict of interest statement in the “Confidential to Editor” section, and submit your "Accept" recommendation.

Reviewer #1: (No Response)

Reviewer #2: All comments have been addressed

2. Is the manuscript technically sound, and do the data support the conclusions?

Reviewer #1: Yes

Reviewer #2: (No Response)

3. Has the statistical analysis been performed appropriately and rigorously? 

Reviewer #1: Yes

Reviewer #2: (No Response)

4. Have the authors made all data underlying the findings in their manuscript fully available?

Reviewer #1: Yes

Reviewer #2: (No Response)

5. Is the manuscript presented in an intelligible fashion and written in standard English?

Reviewer #1: Yes

Reviewer #2: (No Response)

6. Review Comments to the Author

Reviewer #1: Issue 1: Figure 1, the sub-panel label “g” is absent.

Issue 2: Figure 2d, no raw data.

Issue 3: The raw data sheet “Figure2 i—I” should be corrected to “Figure2 i—l”.

Issue 4: Many histograms/panels have been changed with different data or numbers. Please clarify or explain these changes clearly. For example, in figure 2n, all the numbers appear to have changed. The most obvious changes are in the PC and control group. The PC group has a level greater than 0.5 with a p<0.05, whereas in the original version, this number was less than 0.5 with a p<0.01. Similarly, the control group also shows a higher value compared to the original version. What are the reasons for these changes?

Issue 5: Similar issues occur for other panels in other figures such as figure 2l (ftl1 mRNA) PC group, figure 2o (trf protein) all groups, figure 2p (steap3 protein) all groups, figure 2q (ftl protein) all groups, and figure 3j (gpx4 protein) PC group. Please clarify or explain these changes clearly.

Issue 6: Several histograms/panels have updated different numbers and P value labels in Figures 4-7, compared to the original version. Please clarify or explain these changes clearly for each sub-figure that has changed.

Issue 7: Figure 4K, the bands in the TRF raw image do not match the bands in figure 4K. Please also update the raw images to correspond to the TRF bands in figure 4K.

Issue 8: Similar issues occur for figure 5g; all the bands do not match the raw images well, especially the SLC7A11 and beta-actin raw images. Did the authors mistakenly shorten the images horizontally? All the bands appear strange compared to the raw images. Please also update the images correctly.

Reviewer #2: (No Response)

7. PLOS authors have the option to publish the peer review history of their article (what does this mean? ). If published, this will include your full peer review and any attached files.

**Do you want your identity to be public for this peer review?** For information about this choice, including consent withdrawal, please see our Privacy Policy .

Reviewer #1: No

Reviewer #2: No

---

## [Author Response · Author response to Decision Letter 2]

10 May 2025

Reviewer #1:

Issue 1: Figure 1, the sub-panel label “g” is absent.

Respond:We added the sub-panel label “g” in figure 1.

Issue 2: Figure 2d, no raw data.

Response: We have added the raw data for Figure 2d.

Issue 3: The raw data sheet “Figure2 i—I” should be corrected to “Figure2 i—l”.

Response: We have corrected “Figure2 i—I” to “Figure2 i—l”.

Issue 4: Many histograms/panels have been changed with different data or numbers. Please clarify or explain these changes clearly. For example, in figure 2n, all the numbers appear to have changed. The most obvious changes are in the PC and control group. The PC group has a level greater than 0.5 with a p<0.05, whereas in the original version, this number was less than 0.5 with a p<0.01. Similarly, the control group also shows a higher value compared to the original version. What are the reasons for these changes?

Response: Thank you for your attention to detail regarding the changes in the histograms and data panels. When uploading the original data, we carefully checked and re-analyzed all the data, which led to some differences from the initial version.

Issue 5: Similar issues occur for other panels in other figures such as figure 2l (ftl1 mRNA) PC group, figure 2o (trf protein) all groups, figure 2p (steap3 protein) all groups, figure 2q (ftl protein) all groups, and figure 3j (gpx4 protein) PC group. Please clarify or explain these changes clearly.

Response: Thank you for your attention to detail regarding the changes in the histograms and data panels. When uploading the original data, we carefully checked and re-analyzed all the data, which led to some differences from the initial version.

Issue 6: Several histograms/panels have updated different numbers and P value labels in Figures 4-7, compared to the original version. Please clarify or explain these changes clearly for each sub-figure that has changed.

Response: Thank you for your attention to detail regarding the changes in the histograms and data panels. When uploading the original data, we carefully checked and re-analyzed all the data, which led to some differences from the initial version.

Issue 7: Figure 4K, the bands in the TRF raw image do not match the bands in figure 4K. Please also update the raw images to correspond to the TRF bands in figure 4K.

Response: We have updated the correct original images to avoid changes caused by incorrect stretching.

Issue 8: Similar issues occur for figure 5g; all the bands do not match the raw images well, especially the SLC7A11 and beta-actin raw images. Did the authors mistakenly shorten the images horizontally? All the bands appear strange compared to the raw images. Please also update the images correctly.

Response: We have updated the correct original images to avoid changes caused by incorrect stretching.

---

## [Decision Letter · Decision Letter 2]

Mechanism of Plantamajoside in inhibiting ferroptosis of pancreatic β cells and treatment of T2DM via activation of the xCT/GPX4 pathway

PONE-D-25-01394R2

Dear Dr. Wen,

We’re pleased to inform you that your manuscript has been judged scientifically suitable for publication and will be formally accepted for publication once it meets all outstanding technical requirements.

Kind regards,

Kai Huang

Academic Editor

PLOS ONE

Additional Editor Comments (optional):

Reviewers' comments:

Reviewer's Responses to Questions

**Comments to the Author**

1. If the authors have adequately addressed your comments raised in a previous round of review and you feel that this manuscript is now acceptable for publication, you may indicate that here to bypass the “Comments to the Author” section, enter your conflict of interest statement in the “Confidential to Editor” section, and submit your "Accept" recommendation.

Reviewer #1: All comments have been addressed

2. Is the manuscript technically sound, and do the data support the conclusions?

Reviewer #1: Yes

3. Has the statistical analysis been performed appropriately and rigorously? 

Reviewer #1: Yes

4. Have the authors made all data underlying the findings in their manuscript fully available?

Reviewer #1: Yes

5. Is the manuscript presented in an intelligible fashion and written in standard English?

Reviewer #1: Yes

6. Review Comments to the Author

Reviewer #1: well done. just make sure both all the raw data and all the original western blot will be updated as supplementary data/files when publish in the near future.

7. PLOS authors have the option to publish the peer review history of their article (what does this mean? ). If published, this will include your full peer review and any attached files.

**Do you want your identity to be public for this peer review?** For information about this choice, including consent withdrawal, please see our Privacy Policy .

Reviewer #1: No

---

## [Editor Report · Acceptance letter]

PONE-D-25-01394R2

PLOS ONE

Dear Dr. Wen,

I'm pleased to inform you that your manuscript has been deemed suitable for publication in PLOS ONE. Congratulations! Your manuscript is now being handed over to our production team.

Kind regards,

on behalf of

Dr. Kai Huang

Academic Editor

PLOS ONE